# The Prevalence and Correlates of Suicidal Ideation, Plans and Suicide Attempts among 15- to 69-Year-Old Persons in Eswatini

**DOI:** 10.3390/bs10110172

**Published:** 2020-11-10

**Authors:** Supa Pengpid, Karl Peltzer

**Affiliations:** 1ASEAN Institute for Health Development, Mahidol University, Salaya, Phutthamonthon, Nakhon Pathom 73170, Thailand; supaprom@yahoo.com; 2Department of Research and Innovation, University of Limpopo, Polokwane, Sovenga 0727, South Africa; 3Department of Psychology, University of the Free State, Bloemfontein 9300, South Africa

**Keywords:** suicidal ideation, suicide attempt, childhood abuse, sexual violence, adolescents, adults, Eswatini

## Abstract

The study aimed to assess the prevalence and associated factors of ever suicide attempt and past 12-month suicidal ideation, plans and/or attempts among persons aged 15–69 years in Eswatini. Cross-sectional nationally representative data from 3281 persons (33 years median age, range 15–69) of the 2014 Eswatini STEPS Survey were analysed. Results indicate that 3.6% of participants had attempted suicide, and 10.1% engaged in past 12-month suicidal ideation, plan and/or attempts. In adjusted logistic regression analysis, having family members who died from suicide and childhood sexual abuse were associated with ever suicide attempt. In addition, in unadjusted analysis, female sex, adult sexual abuse, threats and family member attempted suicide were associated with ever suicide attempt. In adjusted logistic regression, female sex, childhood sexual abuse, adult sexual abuse, threats, family alcohol problems and having family members who died from suicide were associated with past 12-month suicidal ideation, plan and/or attempts. In addition, in unadjusted analysis, 25–34-year-old participants, unemployed and other, childhood physical abuse, violent injury, family member attempted suicide and having had a heart attack, angina or stroke were associated with past 12-month suicidal ideation, plans and/or attempts. One in ten participants were engaged in suicidal ideation, plans and/or attempts in the past 12 months, and several associated factors were identified that can inform intervention programmes.

## 1. Introduction

Suicide can be seen as a continuum, including ”suicidal behaviours: (thinking about suicide, cognitions), plans (propose methods with which to carry out suicide), and attempts (potentially self-injurious conduct with no fatal outcome that may or may not result in injury, with evidence of intentionality of causing death), as well as consummate suicide (self-inflicted death with evidence of intentionality)” [1,2]. With about 800,000 dying from suicide every year, and 79% occurring in low- and middle-income countries, suicide is a global public health issue [3]. For every death by suicide, many more persons attempt suicide [3]. Suicide prevention efforts need to build on the epidemiological profile of suicidal behaviour [4]. Therefore, there is a great need to have national epidemiological population-based data on suicidal behaviour from low- and middle-income countries in Africa, such as Eswatini [4]. This study reported for the first time data on the prevalence and associated factors of suicidal ideation, plans and/or attempts from a nationally representative survey in Eswatini.

Based on data from the World Mental Health Survey, the prevalence of 12-month suicide ideation, plans and attempts were 2.1%, 0.7% and 0.4% in ten low- and middle-income countries [5], including lifetime prevalence of suicide attempt of 0.7% in Nigeria [6], and 2.9% in South Africa [4]. In a community sample from a district in Ethiopia, the 12-month prevalence of suicidal ideation, plans and attempts was 7.0%, 4.6% and 3.7%, respectively [7]. Treatment seeking after suicide attempts (among those with attempt) was 26.0% among community residents in Ethiopia [7]. Globally, among community adult samples, 34%–42% with suicidal thoughts and 49%–55% with suicide attempts received health care [8].

The age-standardized suicide rates for all ages were 16.7 per 100,000 for both sexes (25.4 among males and 9.6 among females) in Eswatini in 2016 and, in comparison, 12.8 for both sexes (21.7 among males and 5.1 among females) in South Africa and 11.4 per 100,000 in lower-middle-income countries [9]. In a national study on school adolescents in 2013 in Eswatini, the prevalence of suicidal ideation (past year) was 17.0% [10], and among attendees of a primary health care centre in Eswatini, 19% reported suicidal ideation [11]. There are no community-based national epidemiological data on nonfatal suicidal behaviour and its correlates in the lower-middle-income country of Eswatini (formerly Swaziland) in Southern Africa. Eswatini has a population of 1.1 million and a life expectancy at birth of 58.6 years [12]. Eswatini has a weak economy, high unemployment, the highest HIV/AIDS prevalence in the world, an uneven distribution of resources, persistent poverty and food insecurity [12].

Risk factors for suicidal ideation, plans and/or attempts may include people with chronic illness, such as epilepsy and cardiovascular disease, personality factors, unemployment and lower socioeconomic status [13,14,15], childhood adversity [16,17], adverse life events [3,18], female sex, younger age and family history of suicide [5,7,19]. Positive mental health has been shown to be protective against suicidal ideation [20]. Moreover, health risk behaviours have been found to be associated with suicidal ideation, plans and/or attempts, including inadequate physical activity and high sedentary behaviour [21,22,23], substance use (alcohol, tobacco, drugs) [24,25,26], passive smoking [27,28], poor dietary behaviour, such as inadequate fruit and vegetable intake [29], and overweight or obesity [17,30]. Several local studies and reports in Eswatini described the problem and possible social determinants of suicidal ideation, plans and/or attempts. According to the Swaziland Ministry of Sports, Culture and Youth Affairs [31], many cases of suicide are linked to intimate partner violence, financial challenges and mental illness [31]. In a qualitative study, suicidal ideation was linked to HIV stigma [32]. In a national household sample of 13–24-year-old females from Swaziland in 2007, the history of sexual violence was associated with suicidal ideation [33]. In a study by the Eswatini Economic Policy Analysis and Research Centre [34] on youth in Eswatini, unemployment, increased idleness, stress, substance use and lack of recreational and structured activities were linked to poor mental health, including suicidal behaviour. According to Mabuza et al. [35], various factors, such as HIV, poverty, urban migration and divorce, contribute to an increase of single parenthood, which negatively affects the psychosocial development of children, placing them at higher risk of suicide attempts.

It was hypothesized that sociodemographic factors, having psychosocial distress, health status and health risk behaviours are associated with suicidal ideation, plans and/or attempts in this African setting in Eswatini. The investigation aimed to estimate for the first time the national prevalence and its correlates of ever suicide attempts and past 12-month suicidal ideation, plans and/or attempts among persons aged 15–69 years in Eswatini.

## 2. Methods

Cross-sectional data from the nationally representative population-based 2014 Eswatini STEPS Survey were analysed [36]. A multi-stage cluster sample design was used to produce representative data for 15–69-year-olds in Eswatini [37]. Stage 1: “All four regions were included as a sampling frame of 216 Primary Sampling Units (PSUs) that were selected using probability proportional to size sampling” [37]. Stage 2: 20 households were selected from each PSU by systematic random sampling [37]. Stage 3: At the household level, one eligible participant was selected by random sampling [37]. The trained national survey team consisting of research and health staff adapted the WHO STEPS survey tools, translated them into siSwati and back into English, piloted them and administered them using a face-to-face structured interview [37]. More information on the sampling and survey data can be publicly accessed; the overall study response rate was 76% [37]. The Swaziland Scientific and Ethics Committee (SEC) (Ref = MH: 599) approved the study, and written informed consent was obtained from the study participants [37].

### 2.1. Measures

Outcome variables included suicidal ideation, suicide plan and suicide attempt in the past 12 months, ever suicide attempt, method of suicide and care seeking, based on the World Health Organization (WHO) STEPwise approach to mental health/suicide surveillance module (details in Appendix A) [37]. Cronbach’s alpha for the suicidal behaviour items (ideation, plans and attempts in the past 12 months) was 0.89 in this sample.

Social and demographic data consisted of work status, education, sex, age and marital status.

Psychosocial distress indicators (based on the STEPS questionnaire) were childhood physical and sexual abuse, adult sexual abuse, threats, family alcohol problems, violent injury and attempt or death by suicide by a family member (details in Appendix A) [37].

Health status and health risk behaviour variables included overweight or obesity (measured Body Mass Index ≥25 kg/m^2^, history of “heart attack or chest pain from heart disease (angina) or a stroke (cerebrovascular accident or incident)”, current tobacco and alcohol use, passive smoking in the past month (closed spaces at work or at home), intake of fruit and vegetables (<5 servings/day) and sedentary behaviour based on the Global Physical Activity Questionnaire (GPAQ) (≥7 h/day) and low physical activity [37]). The GPAQ has been previously validated in nine countries, including Ethiopia and South Africa in Africa, and found to be an acceptable measure of physical activity [38]. Overall, the “STEPS protocols can be utilized to provide aggregate data for valid between-population comparisons” [39].

### 2.2. Data Analysis

Considering the complex study design, statistical analyses were conducted with STATA software version 15.0 (Stata Corporation, College Station, TX, USA). Unadjusted and adjusted logistic regression analyses were utilized to estimate predictors of ever suicide attempt and past 12-month suicidal ideation, plans and/or attempts in the whole sample and in age-stratified (15–34 and 35–69 years) analysis. Variables significant (*p* < 0.05) in univariate analyses were subsequently included in the multivariable models. Only complete cases were included in the analysis. A *p*-level of <0.05 was considered significant.

## 3. Results

### 3.1. Sample and Suicidal Ideation, Plans and/or Attempts Information

The survey sample consisted of 3281 persons (33 years median age, interquartile range 15–69), 54.2% were female, 47.4% had secondary or higher education, 63.8% were never married, separated, divorced or widowed, and 61.2% were employed or were studying. More than one in four participants (27.7%) had experienced childhood physical abuse, 4.7% childhood sexual abuse, 3.0% adult sexual abuse, 13.1% threats in the past 12 months, 2.4% experienced a violent injury in the past 12 months, and 11.3% and 7.6% had a close family member who attempted suicide and who died from suicide, respectively.

More than two in five participants (43.4%) were overweight or obese, 4.4% had had a heart attack, angina or stroke, 7.9% were current tobacco users, 23.4% were exposed to secondary smoke at home or at work, 13.0% current alcohol users, 92.1% consumed <5 servings of fruit and vegetables a day, 6.1% engaged in sedentary behaviour, and 23.9% were physically inactive. More than a hundred participants (3.6%) had attempted suicide, and 10.1% engaged in past 12-month suicidal ideation, plans and/or attempts (see Table 1).

The main methods used in the last suicide attempt were “poisoning with pesticides (e.g., rat poison, insecticide, weed killer)” (41.7%), followed by others (27.8%, mainly hanging with a rope), “overdose of medication (e.g., prescribed, over-the-counter)” (14.8%), “razor, knife or other sharp instrument” (4.3%) and “overdose of other substance (e.g., heroin, crack, alcohol).” About one in four participants (26.1%) sought medical care for the last suicide attempt, of which 43.3% were admitted to the hospital overnight. Of those participants who reported past 12-month suicidal ideation (9.3%), 27.6% had sought professional help.

### 3.2. Associations with Ever Suicide Attempt and Past 12-Month Suicidal Ideation, Plans and/or Attempts

In adjusted logistic regression analysis, childhood sexual abuse and family members who died from suicide were associated with ever suicide attempt. In addition, in unadjusted analysis, female sex, adult sexual abuse, threats and family member attempted suicide were associated with ever suicide attempt.

In adjusted logistic regression, female sex, childhood sexual abuse, adult sexual abuse, threats, death of a family member from suicide and family alcohol problems were associated with past 12-month suicidal ideation, plans and/or attempts. In addition, in unadjusted analysis, 25–34-year-old participants, unemployed and other, childhood physical abuse, violent injury, family member attempted suicide and having had a heart attack, angina or stroke were associated with past 12-month suicidal ideation, plans and/or attempts (see Table 2).

In age-stratified multivariable logistic regression analysis, among 15–34-year-old participants, childhood sexual abuse and having a family member who died from suicide, and among 35–69-year-old participants, none of the study variables were associated with ever having attempted suicide. Among 15–34-year-old participants, female sex, childhood sexual abuse, adult sexual abuse and threats were associated with past 12-month suicidal behaviour, while among 35–69-year-old respondents, threats and inadequate intake of fruit and vegetables were associated with past 12-month suicidal behaviour (see Table 3).

## 4. Discussion

The investigation aimed to estimate the prevalence and correlates of ever suicide attempt and past 12-month suicidal ideation, plans and/or attempts among 15–69-year-old persons in Eswatini. The prevalence of lifetime suicide attempts (3.6%) was higher than in previous studies in Nigeria (0.7%) [6], South Africa (2.9%) [4] and Bhutan (0.7%) [19], and the prevalence of past 12-month suicidal ideations (9.3%), plan (5.4%) and attempts (2.1%) was higher than in ten low- and middle-income countries (ideation 2.1%, plan 0.7% and attempts 0.4%) [5] but similar to a low-resourced community in Ethiopia (ideation 7.0%, plan 4.6% and attempts 3.7%) [7]. The higher rate of suicidal ideation, plans and/or attempts in Eswatini than in many other countries was also confirmed in a previous study among adolescents in Eswatini (17.0% past 12-month suicidal ideation) [10] and in a study among postnatal health care attendees (19% suicidal ideation) in Eswatini [11], and the higher suicide rate of 16.7 per 100,000 in Eswatini compared to lower-middle-income countries (11.4 per 100,000) [9].

In this study, we found that female sex, childhood sexual abuse, adult sexual abuse, threats, death of a family member from suicide and family alcohol problems increased the odds for past 12-month suicidal ideation, plans and/or attempts and/or ever suicide attempt. The main methods used in the last suicide attempt were poisoning with pesticides, hanging with a rope and overdose of medication, which concurs with the finding of high rates of suicides due to pesticide self-poisoning, hanging and firearms, in particular in low- and middle-income countries [3]. Knowing the specific suicide methods can help in designing suicide prevention strategies by restricting access to means of suicide [3]. In this study, of those participants who reported past 12-month suicidal ideation, 27.6% had sought professional help, which is lower than globally among community adult samples (34%-42%) [8], and among participants with a suicide attempt, 26.1% sought medical care which is lower than globally among community adult samples (49% to 55%) [8].

The very low uptake of health care seeking in this study may be related to the stigma that is associated with suicidality and unavailable mental health services [7]. The Eswatini Ministry of Health recognizes that there is evidence of high rates of suicide that are not appropriately addressed [40]. Not having a mental health policy and suicide prevention strategy [41], the Eswatini national health sector Strategic Plan proposes to integrate mental health care, rehabilitation and counselling into service delivery by packaging and decentralizing mental health services at all levels, develop rehabilitation centres for substance abuse and strengthen empowerment approaches for mental health clients (e.g., support groups) [40]. To support this, the Eswatini Ministry of Health and others developed a brief psychological intervention for common mental disorders suitable for the implementation in primary care in Eswatini [42]. Apart from primary health care screening for depression and suicidal ideation, for people not consulting primary care, community-level case-finding strategies may increase detection and management [5,43].

Consistent with previous research [5,7,19], this study found that female sex and, in unadjusted analysis, 25–34-year-olds and being unemployed were associated with suicidal ideation, plans and/or attempts. In contrast to international trends where suicide attempts are higher among women than men, in a review on studies in Africa, mixed results were found, with some studies reporting male or female predominance, but other studies did not find any sex differences [44]. We did not find significant differences in the prevalence of suicidal ideation, plans and/or attempts in terms of educational level and marital status, while some previous studies [5,13,19] showed associations between lower socioeconomic or educational status and being single, separated, divorced or widowed and suicidal ideation, plans and/or attempts. In line with former research findings on the family history of suicide [5,7,19], this study found an association between the death of a family member from suicide and, in unadjusted analysis, a family member’s attempted suicide and suicidal ideation, plans and/or attempts. In this study, a high proportion of participants reported having a family member who had attempted suicide (11.3%) and a family member who had died from suicide (7.6%). Various guidelines and suicide postvention service models have been developed and systematically reviewed showing that little evidence of effectiveness exists. However, the involvement of trained volunteers/peers in grief counselling seems to be a promising postvention [45,46].

In agreement with previous investigations [3,16,18,47], this study showed that childhood adversity (childhood sexual abuse and, in unadjusted analysis, childhood physical abuse) and adverse life events (adult sexual abuse, threats, family alcohol problems and, in unadjusted analysis, violent injury) increased the likelihood of past 12-month suicidal ideation, plans and/or attempts and/or ever suicide attempt. The rates of childhood sexual abuse, adult sexual abuse and suicidal ideation, plans and/or attempts were high in this study, with about one-third of all participants with suicidal ideation, plans and/or attempts experiencing childhood and/or adult sexual abuse. In a community survey sample of girls and women aged 13–24 years in Eswatini in 2007, a high proportion (33.2%) reported childhood sexual violence, with one of the consequences being self-reported depression and suicidal ideation [32]. Possible mechanisms that link child sexual abuse with adult suicidal ideation, plans and/or attempts include psychosocial development, psychiatric and health functioning and biological factors [48]. For individuals who have experienced childhood maltreatment such as sexual abuse, early interventions to reduce suicide risk may be beneficial [16]. These could include periodic screening for experiences of childhood abuse among persons with suicidal behaviour, followed by therapeutic management [16].

Furthermore, the study showed that family alcohol problems increased the odds of suicidal ideation, plans and/or attempts. The World Health Organization estimates that alcohol significantly contributes to suicide, and after alcohol consumption, the risk of a suicide attempt increases manifold [49]. The prevalence of heavy episodic drinking in Eswatini in 2016 was high, with 17.3% in the general population (≥15 years) and 60% among drinkers [50], and the prevalence of alcohol use disorders among both sexes was 7.1%, higher than 3.7% in the WHO Africa region [50].

In unadjusted analysis, having had a heart attack, angina or stroke (cardiovascular disease) increased the odds of suicidal ideation, plans and/or attempts. Similar results have been found in previous research [15,51]. It is possible that an increase in disability in cardiovascular disease patients leads to an increase in suicidal ideation, plans and/or attempts [51,52]. Among 35–69-year-old participants in this study, an inadequate fruit and vegetable intake was significantly positively associated with past 12-month suicidal behaviour, which is consistent with a study among adults in Japan [29]. The potentially beneficial effects of fruit and vegetable consumption against depression have been demonstrated in various studies [53]. Unlike some previous research findings [17,21,22,23,24,25,26,27,28,30], this investigation did not show a significant association between overweight or obesity, current tobacco use, current alcohol use, inadequate physical activity, sedentary behaviour, passive smoking and suicidal ideation, plans and/or attempts.

The study limitations included that this investigation was limited due to the self-report of most data and the cross-sectional survey design. An additional limitation was that the 2014 Eswatini STEPS Survey did not assess other mental disorders, such as depression and parental psychopathology. Since some variables, such as household income, had too many missing values, they were excluded from the analysis.

## 5. Conclusions

The results of this nationally representative population survey of persons aged 15–69 years in Eswatini showed that one in ten participants had engaged in suicidal ideation, plans and/or attempts in the past 12 months, and 3.6% had attempted suicide. Several risk factors for past 12-month suicidal ideation, plans and/or attempts and/or ever suicide attempt were identified, including female sex, childhood sexual abuse, adult sexual abuse, threats, having family members who died from suicide and family alcohol problems, which can help in the design of strategies to prevent suicidal ideation, plans and/or attempts in Eswatini.

## Figures and Tables

**Table 1 behavsci-10-00172-t001:** Sample and suicidal behaviour characteristics among 15–69-year-old persons in Eswatini.

Variable	Sample	Ever Suicide Attempt	Suicidal Behaviour ^1^
	N (%)	N (%)	N (%)
**Socio-demographic factors**			
**All**	3281	115 (3.6)	339 (10.1)
Age (years)			
50–69	724 (22.1)	21 (2.7)	63 (8.9)
35–49	838 (25.5)	34 (3.1)	87 (8.9)
25–34	788 (24.0)	31 (4.9)	104 (14.0)
15–24	931 (28.4)	29 (3.3)	85 (9.0)
Gender			
Male	1137 (45.8)	26 (2.4)	76 (6.7)
Female	2144 (54.2)	89 (4.6)	263 (13.1)
Education			
<Primary	1026 (24.3)	34 (3.3)	109 (9.4)
Primary	831 (28.3)	38 (3.6)	101 (10.8)
Secondary and higher	1421 (47.4)	43 (3.7)	128 (10.1)
Marital status			
Married/cohabiting	1485 (36.2)	58 (3.9)	169 (11.4)
Single/divorced/separated/widowed	1793 (63.8)	57 (3.4)	169 (9.4)
Work status			
Employed/student	1773 (61.2)	53 (2.9)	155 (8.6)
Unemployed and other	1505 (38.8)	62 (4.6)	184 (12.6)
**Psychosocial distress**			
Childhood physical abuse	935 (27.7)	39 (4.2)	125 (13.6)
Childhood sexual abuse	185 (4.7)	23 (15.0)	47 (31.8)
Adult sexual abuse	118 (3.0)	13 (14.2)	31 (35.1)
Threats	408 (13.1)	27 (6.4)	86 (19.3)
Alcohol family problem	252 (7.9)	12 (4.8)	46 (18.2)
Violent injury	74 (2.4)	6 (7.4)	18 (19.0)
Family member attempted suicide	341 (11.3)	18 (6.7)	56 (17.0)
Family member died from suicide	247 (7.6)	15 (8.9)	41 (19.3)
**Health status and risk behaviours**			
Overweight or obese	1485 (43.4)	66 (4.7)	159 (11.0)
Heart attack, angina or stroke	133 (4.4)	8 (6.3)	28 (16.0)
Current tobacco use	272 (7.9)	8 (2.8)	35 (13.8)
Current alcohol use	403 (13.0)	13 (2.8)	46 (10.9)
Inadequate fruit and vegetable intake	3013 (92.1)	107 (3.5)	316 (10.3)
Inadequate physical activity	819 (23.9)	30 (3.6)	89 (11.2)
Sedentary behaviour	174 (6.1)	7 (3.2)	22 (11.8)
Passive smoking	685 (23.4)	26 (3.2)	91 (12.4)

^1^ Past 12-month suicidal ideation (9.3%) and/or suicide plan (5.4%) and/or suicide attempt (2.1%).

**Table 2 behavsci-10-00172-t002:** Associations with ever suicide attempt and past 12-month suicidal behaviour.

Variable	Ever Suicide Attempt		Suicidal Behaviour (Past 12 Months) ^1^	
	COR (95% CI)	AOR (95% CI)	COR (95% CI)	COR (95% CI)
**Socio-demographic factors**				
Age (years)				
50–69	1 (Reference)	---	1 (Reference)	1 (Reference)
35–49	1.15 (0.54, 2.45)		0.99 (0.59, 1.67)	1.03 (0.59, 1.78)
25–34	1.82 (0.78, 4.26)		1.66 (1.06, 2.60) *	1.58 (1.00, 2.51)
15–24	1.23 (0.57, 2.63)		1.00 (0.61, 1.64)	1.16 (0.68, 1.97)
Gender				
Male	1 (Reference)	1 (Reference)	1 (Reference)	1 (Reference)
Female	1.96 (1.11, 3.48) *	1.63 (0.91, 2.93)	2.11 (1.49, 2.98) ***	1.80 (1.24, 2.60) **
Education				
<Primary	1 (Reference)	---	1 (Reference)	--
Primary	1.08 (0.59, 1.96)		1.16 (0.80, 1.69)	
Secondary and higher	1.11 (0.60, 2.05)		1.08 (0.76, 1.54)	
Marital status				
Married/cohabiting	1 (Reference)	---	1 (Reference)	---
Single/divorced/separated/widowed	0.85 (0.52, 1.38)		0.81 (0.59, 1.11)	
Work status				
Employed/student	1 (Reference)	---	1 (Reference)	1 (Reference)
Unemployed and other	1.58 (0.95, 2.62)		1.54 (1.16, 2.05) **	1.24 (0.92, 1.66)
**Psychosocial distress**				
Childhood physical abuse	1.27 (0.73, 2.18)	---	1.62 (1.16, 2.27) **	1.25 (0.89, 1.77)
Childhood sexual abuse	5.68 (3.22, 10.01) ***	3.70 (1.80, 7.62) ***	4.67 (3.03, 7.21) ***	2.10 (1.22, 3.63) **
Adult sexual abuse	4.92 (2.60, 9.32) ***	1.02 (0.94, 3.95)	5.22 (2.96, 9.21) ***	2.93 (1.57, 3.63) ***
Threats	2.08 (1.16, 3.76) *	1.75 (0.95, 3.23)	2.50 (1.78, 3.53) ***	1.95 (1.36, 2.82) ***
Alcohol family problem	1.39 (0.63, 3.06)	---	2.13 (1.36, 3.34) ***	1.77 (1.10, 2.83) *
Violent injury	2.23 (0.83, 5.99)	---	2.13 (1.22, 3.69) **	1.25 (0.66, 2.37)
Family member attempted suicide	2.17 (1.04, 4.50) *	1.04 (0.51, 2.12)	2.05 (1.37, 3.07) ***	1.29 (0.80, 2.08)
Family member died from suicide	3.02 (1.41, 6.47) **	2.66 (1.28, 5.51) **	2.30 (1.43, 3.71) ***	1.84 (1.01, 3.36) *
**Health status and risk behaviours**				
Overweight or obese	1.60 (0.97, 2.65)	---	1.20 (0.88, 1.64)	---
Heart attack, angina or stroke	1.88 (0.70, 5.06)	---	1.74 (1.02, 2.97) *	1.28 (0.68, 2.40)
Current tobacco use	0.76 (0.29, 1.97)	---	1.47 (0.88, 2.44)	---
Current alcohol use	0.76 (0.39, 1.49)	---	1.09 (0.71, 1.68)	---
Intake of fruit and vegetables (<5 servings/day)	0.85 (0.37, 1.97)	---	1.33 (0.73, 2.42)	---
Inadequate physical activity	1.03 (0.58, 1.84)	---	1.14 (0.80, 1.61)	---
Sedentary behaviour	0.87 (0.31, 2.46)	---	1.16 (0.69, 1.96)	---
Passive smoking	0.87 (0.49, 1.57)	---	1.35 (0.97, 1.88)	---

^1^ Past 12-month suicidal ideation (9.3%) and/or suicide plan (5.4%) and/or suicide attempt (2.1%); COR = crude odds ratio; AOR = adjusted odds ratio; *** *p* < 0.001, ** *p* < 0.01, * *p* < 0.05.

**Table 3 behavsci-10-00172-t003:** Age-stratified associations with ever suicide attempt and past 12-month suicidal behaviour.

Variable	Ever Suicide Attempt		Suicidal Behaviour (Past 12 Months) ^1^	
	15–34 Years	35–69 Years	15–34 Years	35–69 Years
	AOR^2^ (95% CI)	AOR^2^ (95% CI)	AOR^2^ (95% CI)	AOR^2^ (95% CI)
**Socio-demographic factors**				
Gender				
Male	1 (Reference)	---	1 (Reference)	---
Female	1.30 (0.54, 3.14)		1.70 (1.01, 2.87) *	
Education				
<Primary	---	---	---	---
Primary				
Secondary and higher				
Marital status	---	---		---
Married/cohabiting			1 (Reference)	
Single/divorced/separated/widowed			0.82 (0.52, 1.29)	
Work status		---		---
Employed/student	1 (Reference)		1 (Reference)	
Unemployed and other	1.54 (0.79, 3.00)		1.26 (0.82, 1.94)	
**Psychosocial distress**				
Childhood physical abuse	---	---	1.31 (0.81, 2.11)	---
Childhood sexual abuse	5.07 (1.88, 13.68) ***	2.15 (0.94, 4.92)	3.31 (1.64, 6.66) ***	---
Adult sexual abuse	1.36 (0.46, 4.01)	2.39 (0.83, 6.88)	4.18 (1.74, 10.02) ***	---
Threats	1.71 (0.66, 4.46)	---	2.16 (1.24, 3.77) **	2.18 (1.30, 3.65) **
Alcohol family problem	---	---	1.82 (0.93, 360)	1.75 (0.84, 3.65)
Violent injury	1.62 (0.65, 4.02)	---	1.13 (0.57, 2.24)	---
Family member attempted suicide	---	---	---	1.67 (0.73, 3.85)
Family member died from suicide	3.30 (1.05, 10.33) *	---	---	1.48 (0.48, 4.55)
**Health status and risk behaviours**				
Overweight or obese	1.68 (0.81, 3.50)	---	1.42 (0.92, 2.18)	0.64 (0.40, 1.03)
Heart attack, angina or stroke	1.85 (0.59, 5.80)	---	---	---
Current tobacco use	---	---	---	1.60 (0.83, 3.06)
Current alcohol use	---	---	---	---
Intake of fruit and vegetables (<5 servings/day)	---	---	---	4.12 (1.24, 13.70) *
Inadequate physical activity	---	---	---	---
Sedentary behaviour	---	---	---	---
Passive smoking	---	---	---	---

^1^ Past 12-month suicidal ideation (9.3%) and/or suicide plan (5.4%) and/or suicide attempt (2.1%); COR = crude odds ratio; AOR = adjusted odds ratio; ^2^ Variables significant in univariate analysis were included in the multivariable model; *** *p* < 0.001, ** *p* < 0.01, * *p* < 0.05.

## Data Availability

The data for the current study are publicly available at the World Health Organization NCD Microdata Repository (URL: https://extranet.who.int/ncdsmicrodata/index.php/catalog).

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
