# Peer review of "The Prevalence and Correlates of Suicidal Ideation, Plans and Suicide Attempts among 15- to 69-Year-Old Persons in Eswatini"

_behavsci, 2020, doi:10.3390/bs10110172_

Round 1

Reviewer 1 Report

Very short introduction and the theoretical review needs to be improved. There are studies of interest that are not mentioned. For example:

https://www.mdpi.com/1660-4601/16/22/4496 

https://doi.org/10.1016/j.ijchp.2020.03.005 

http://dx.doi.org/10.5093/ejpalc2018a2 

https://doi.org/10.1016/j.ijchp.2019.02.003 

https://doi.org/10.1016/j.ijchp.2017.08.001 

The source of “2014 Eswatini STEPS” is unclear. There is no correspondence with the reference.

The data are from 2014. Justifying why are not analyzed more recent data. Information for 2018 (published in 2019) are available:

https://www.uneca.org/sites/default/files/uploaded-documents/STEPS/2018/eswatini-country-profile.pdf  

Does the Questionnaire have adequate psychometric properties?  Must be indicated.

Information on the availability of data and materials must be in the "method".

Author Response

Reviewer I:
Very short introduction and the theoretical review needs to be improved. There are studies of interest that are not mentioned. For example:
https://www.mdpi.com/1660-4601/16/22/4496
Mejías-Martín, Y.; Luna del Castillo, J.D.; Rodríguez-Mejías, C.; Martí-García, C.; Valencia-Quintero, J.P.; García-Caro, M.P. Factors Associated with Suicide Attempts and Suicides in the General Population of Andalusia (Spain). Int. J. Environ. Res. Public Health 2019, 16, 4496.
https://doi.org/10.1016/j.ijchp.2020.03.005
Lew B, Osman A, Huen JMY, Siau CS, Talib MA, Cunxian J, Chan CMH, Leung ANM. A comparison between American and Chinese college students on suicide-related behavior parameters. Int J Clin Health Psychol. 2020 May-Aug;20(2):108-117. doi: 10.1016/j.ijchp.2020.03.005. Epub 2020 May 12. PMID: 32550850; PMCID: PMC7296251.
http://dx.doi.org/10.5093/ejpalc2018a2
https://doi.org/10.1016/j.ijchp.2019.02.003
https://doi.org/10.1016/j.ijchp.2017.08.001
Response: more of the latter 3 references has been included in the introduction
The source of “2014 Eswatini STEPS” is unclear. There is no correspondence with the reference.
Response: the URL to the reference [32] has been updated
Cross-sectional data from the nationally representative “2014 Eswatini STEPS Survey” [32] were analyzed.
The data are from 2014. Justifying why are not analyzed more recent data. Information for 2018 (published in 2019) are available:
https://www.uneca.org/sites/default/files/uploaded-documents/STEPS/2018/eswatini-country-profile.pdf
Response: The above reference refers to “Structural transformation, employment, production and society” STEPS and has no information on suicidal behaviour at all
Does the Questionnaire have adequate psychometric properties? Must be indicated.
Response: more is added, as in below
Outcome variables included suicidal ideation, suicide plan and suicide attempt in the past 12 months, ever suicide attempt, method of suicide and care seeking, based on the “World Health Organization (WHO) STEPwise approach to surveillance- Mental Health-Suicide module (details in Supplementary file 1) [37]. Cronbach alpha for the suicidal behaviour items (ideation, plans and attempts in the past 12 months) was 0.89 in this sample.
Social and demographic data consisted of work status, education, sex, age, and marital status.
Psychosocial distress indicators (based on the STEPS questionnaire) were childhood physical and sexual abuse, adult sexual abuse, threats, alcohol family problems, violent injury, attempt or death by suicide by a family member (details in Supplementary file 1) [37].
Health status and health risk behaviour variables included overweight or obesity (measured “Body Mass Index” ≥25kg/m2, history of “heart attack or chest pain from heart disease (angina) or a stroke (cerebrovascular accident or incident)”, current tobacco and alcohol use, passive smoking in the past month (closed spaces at work or at home), intake of fruit and vegetables (<5 servings/day), and based on the “Global Physical Activity Questionnaire” (GPAQ) sedentary behaviour (≥7 hours/day) and low physical activity [37]. The GPAQ has been previously validated in nine countries, including Ethiopia and South Africa in Africa, and found an acceptable measure of physical activity [38]. Overall, the “STEPS protocols can be utilized to provide aggregate data for valid between-population comparisons.” [39].

Information on the availability of data and materials must be in the "method".
Response: below is in the method section
More information on the sampling and survey data can be publicly accessed; the overall study response rate was 76%.”[32,33]

Reviewer 2 Report

This paper describes an empirical study to assess the prevalence and associated factors of suicide attempt and past month suicidal ideation, plans, or attempts. Considering the high rates of suicide worldwide and the lack of studies in the community population, this study's subject is relevant. It may be useful for mental health professionals. However, there are several aspects that have not been sufficiently explained or justified in this article, and there are some aspects of the introduction, procedure, results and discussion that should be reviewed, as specified below.

I have a few critiques for the authors' consideration to improve the article.

  1. Reading the introduction is difficult. Please consider using linguistic connectors to join parts of the manuscript and establish logical relationships between sentences. In this way, you can present the background and establish a common thread.
  2. Although the authors have made an effort to present the antecedents in the population under study, the authors must present data from similar studies carried out in other countries.
  3. The document lacks the "participants" section. Please, add the section and explain the characteristics of your sample, found in the results.
  4. The data was extracted from a national survey from 2014. There are no more recent data? The situations of the population can change in 6 years.
  5. The manuscript does not specify which instrument the authors used to measure psychosocial distress. Please provide the information.
  6. The authors have a sufficient sample to separate the study sample by age group. These analyzes would provide information on the population at risk in the population under study.
  7. Regarding the discussion, there is no relationship with the introduction. In fact, the authors do not relate the data set out in the introduction with their data.
  8. There is consensus in the literature on the predictive value of depression on different suicidal manifestations, including suicide risk, the authors did not take into account this variable, which results should be interpreted with caution, because many variables related to risk are no longer significant when controlling depression, because this variable is one of the main predictors of suicidal risk.

Author Response

Reviewer II
This paper describes an empirical study to assess the prevalence and associated factors of suicide attempt and past month suicidal ideation, plans, or attempts. Considering the high rates of suicide worldwide and the lack of studies in the community population, this study's subject is relevant. It may be useful for mental health professionals. However, there are several aspects that have not been sufficiently explained or justified in this article, and there are some aspects of the introduction, procedure, results and discussion that should be reviewed, as specified below.
I have a few critiques for the authors' consideration to improve the article.
1. Reading the introduction is difficult. Please consider using linguistic connectors to join parts of the manuscript and establish logical relationships between sentences. In this way, you can present the background and establish a common thread.
Response: corrected
2. Although the authors have made an effort to present the antecedents in the population under study, the authors must present data from similar studies carried out in other countries.
Response: below is reported
Based on data from the “World Mental Health Survey”, the prevalence of 12-month suicide ideation, plans and attempts were 2.1%, 0.7% and 0.4% in ten low- and middle-income countries [3], including lifetime prevalence of suicide attempt of 0.7% in Nigeria [4], and 2.9% in South Africa [2]. In a community sample from a district in Ethiopia, the 12-month prevalence of suicidal ideation, plans and attempts was 7.0%, 4.6%, and 3.7%, respectively [5].
3. The document lacks the "participants" section. Please, add the section and explain the characteristics of your sample, found in the results.

Response: below is described

Sample and suicidal ideation, plans and/or attempts information
The survey sample consisted of 3,281 persons (33 years median age, 25 IQR, range 15-69), 54.2% were female, 47.4% had secondary or higher education, 63.8% were never married, separated, divorced or widowed and 61.2% were employed or were studying. More than one in four participants (27.7%) had experienced childhood physical abuse, 4.7% childhood sexual abuse, 3.0% adult sexual abuse, 13.1% threats in the past 12 months, 2.4% experienced violent injury in the past 12 months, 11.3% and 7.6% had a close family member who attempted suicide and who died from suicide, respectively [33].
4. The data was extracted from a national survey from 2014. There are no more recent data? The situations of the population can change in 6 years.
Response: No

5. The manuscript does not specify which instrument the authors used to measure psychosocial distress. Please provide the information.
Response: below is added
Psychosocial distress indicators (based on the STEPS questionnaire) were childhood physical and sexual abuse, adult sexual abuse, threats, alcohol family problems, violent injury, attempt or death by suicide by a family member (details in Supplementary file 1) [33].
6. The authors have a sufficient sample to separate the study sample by age group. These analyzes would provide information on the population at risk in the population under study.
Response: age stratified models are added

In age stratified multivariable logistic regression analysis, among 15-34 year-old participants, childhood sexual abuse and family member died from suicide, and among 35-69 year-old participants none of the study variables were associated with ever having attempted suicide. Among 15-34 year-old participants, female sex, childhood sexual abuse, adult sexual abuse and threats were associated with past 12-month suicidal behaviour, while among 35-69 year old respondents threats and inadequate intake of fruit and vegetables were associated with past 12-month suicidal behaviour (see Table 3).

7. Regarding the discussion, there is no relationship with the introduction. In fact, the authors do not relate the data set out in the introduction with their data.
Response: This is not correct
8. There is consensus in the literature on the predictive value of depression on different suicidal manifestations, including suicide risk, the authors did not take into account this variable, which results should be interpreted with caution, because many variables related to risk are no longer significant when controlling depression, because this variable is one of the main predictors of suicidal risk.
Response: depression was not assessed in this study, and has been added under study limitations

Reviewer 3 Report

Reviewer comments on “Prevalence and correlates of suicidal ideation …”

This is a promising article, which deserves publication following revision.

First, although the authors can justify the term “attempted suicide” by reference to the quoted WHO definition in reference [1], some controversies surrounding this definition should be clarified.

Most “suicide attempts” are better termed “suicidal gestures”, in which the act of self-harm is a ‘cry for help’ rather than a failed attempt  to take one’s life. Completed suicide is a very rare phenomenon; but suicidal gestures are common, and the large majority of these gestures do not carry any risk of fatality. They do however constitute an important reason for intervention, as this paper makes clear in referring to the publications [39] and [40]. These definitional issues should be discussed by pointing out that being female is one of the risk factors for so-called “suicide attempts”, while being male is a marker for actually completing suicide. This differential sex ratio should be discussed.

Secondly, there is insufficient information about the questionnaire used, sample selection, and how the interview was administered, and by whom. I followed up the references given – [32] and [33] – but these do not list the actual wording of the parts of the questionnaire regarding social stress, self-harm,  and mental health outcomes. The reports of many countries are indexed in the WHO reference, but Swaziland/Eswatini is not among them. Fuller information (and better referencing) on the “WHO STEPWISE Approach to Surveillance Instrument” must be given. In what language was the questionnaire completed by respondents? Is Eswatini a multilingual nation?

It is suggested that adding “Childhood Abuse” and “Sexual Violence” to Keywords would be valuable. Overall, the review of literature on suicidal behaviours in Africa is comprehensive and well-done.

Author Response

This is a promising article, which deserves publication following revision.
First, although the authors can justify the term “attempted suicide” by reference to the quoted WHO definition in reference [1], some controversies surrounding this definition should be clarified.
Most “suicide attempts” are better termed “suicidal gestures”, in which the act of self-harm is a ‘cry for help’ rather than a failed attempt to take one’s life. Completed suicide is a very rare phenomenon; but suicidal gestures are common, and the large majority of these gestures do not carry any risk of fatality. They do however constitute an important reason for intervention, as this paper makes clear in referring to the publications [39] and [40]. These definitional issues should be discussed by pointing out that being female is one of the risk factors for so-called “suicide attempts”, while being male is a marker for actually completing suicide. This differential sex ratio should be discussed.
Response: As discussed in below, there is no clear differential sex ratio in Africa
In contrast to international trends where suicide attempts are higher in women than in men, in a review on studies in Africa, mixed results were found, with some studies reporting male or female predominance, but other studies did not find sex differences [38].
Secondly, there is insufficient information about the questionnaire used, sample selection, and how the interview was administered, and by whom. I followed up the references given – [32] and [33] – but these do not list the actual wording of the parts of the questionnaire regarding social stress, self-harm, and mental health outcomes. The reports of many countries are indexed in the WHO reference, but Swaziland/Eswatini is not among them. Fuller information (and better referencing) on the “WHO STEPWISE Approach to Surveillance Instrument” must be given. In what language was the questionnaire completed by respondents? Is Eswatini a multilingual nation?
Response:
The actual questionnaire is provided in the supplementary file, it is also available in the references given.
Further information on the sampling, administration and translation of questionnaires is added as in below
A multi-stage cluster sample design was used to produce representative data for 15-69 year-olds in Eswatini [33] Stage 1: “All four regions were included as a sampling frame of 216 Primary Sampling Units (PSUs) that were selected using probability proportional to size sampling” [33]. Stage 2: 20 households were selected from each PSU by systematic random sampling [33]. Stage 3: At household level one eligible participant was selected by random sampling [33]. The trained national survey team consisting of research and health staff adapted the WHOSTEPS survey tools, translated them into siSwati and back translated into English, piloted them and admistered them using a face-to-face structured interview [33].
It is suggested that adding “Childhood Abuse” and “Sexual Violence” to Keywords would be valuable.
Response: added
Overall, the review of literature on suicidal behaviours in Africa is comprehensive and well-done.

Round 2

Reviewer 2 Report

I am satisfied with the changes made. However, I do have some concerns.

  1. The data presented by the authors corresponds to a too old national survey (2014). Therefore, the data is not new, or various factors may have changed.

  2. The manuscript is disorganized. Information that must be included in the results method is presented.

  3. The instruments used have questionable psychometric properties. No data are presented that allow assessing reliability and validity.